# Understanding Suicide Stigma in Fly-In/Fly-Out Workers: A Thematic Analysis of Attitudes Towards Suicide, Help-Seeking and Help-Offering

**DOI:** 10.3390/ijerph22030395

**Published:** 2025-03-07

**Authors:** Jordan Jackson, Victoria Ross

**Affiliations:** Australian Institute for Suicide Research and Prevention, School of Applied Psychology, Griffith University, Brisbane, QLD 4122, Australia; jordan.jackson@griffithuni.edu.au

**Keywords:** fly-in/fly out, suicide, stigma, mental health, workplace

## Abstract

Background: Suicide is estimated to be the fourth leading cause of death globally, with those working in male-dominated industries such as mining and construction at higher risk than the general population. Research suggests this is due (in part) to stigma towards mental health. No research exists that has sought to understand the attitudes underpinning this stigma in the fly-in/fly-out (FIFO) industry. The current study, set in Australia, is the first of its kind to examine what specific stigmatised attitudes of FIFO workers exist towards suicide, help-seeking, and help-offering. Methods: Using convenience sampling, FIFO workers (*n* = 138) completed an online self-report survey. General thematic analysis identified four major themes. Most salient was that fear of negative consequences for employment was a primary barrier to help-seeking and help-offering. Participants also expressed lack of trust in leadership and workplace mental health culture, lack of knowledge and confidence in responding to suicidality disclosure, and fear of negative reactions as barriers to help-seeking and help-offering behaviours. Conclusions: These findings present new and valuable insights into why FIFO workers are reluctant to seek or offer help for suicidality and have important implications for addressing systematic inadequacies within the sector that hinder disclosure of suicidal ideation and access to vital services.

## 1. Introduction

Recent WHO figures estimate suicide to be the fourth leading cause of death globally, with over 700,000 persons dying by suicide every year [1]. The age-standardised suicide rates in Australia shows that the male suicide rate is approximately three times greater than that of females, with this trend remaining stable over the last ten years [2]. This comparatively higher rate of male suicides is consistent internationally, giving rise to a multitude of male-centric suicide prevention initiatives [3]. Once such area of focus has centred around the male-dominated industry of resources mining [4] and the sub-industry of fly-in/fly-out workers [5].

### 1.1. Fly-In/Fly-Out (FIFO) Workers

Fly-in/fly-out (FIFO) work is a common phrase used to describe an occupational methodology involving workers flying from their place of residence to a workplace, residing in accommodation on that workplace for an extended period while they work every day, before flying home to their place of residence for a period of respite [5]. FIFO work is traditionally associated with the resources industry (e.g., mineral, oil, and gas mining) [6,7].

It is estimated that the prevalence of suicide amongst male mining workers is likely 25 per 100,000 and increasing [8]. FIFO work has shown to have a negative impact on the mental health of workers for a number of reasons, including the specific requirements of the job (e.g., long working days and adverse environmental conditions) [7]; a lack of autonomy [9]; long periods of time away from family, friends, and support networks (i.e., isolation) [10]; and the consistent strain of having to maintain two separate lives of home and work simultaneously [5,10]. While these factors are significant contributors, it has been argued that they could be mitigated with targeted interventions [7,10,11]. However, mental health stigma permeates the industry and has been identified as a barrier that makes intervention more complicated [7,10,11]. Current research suggests that the stigma around mental health in male-dominated industries stems from traditional masculine norms [5,10,12]; however, research into what specific attitudes underpin the broad term of ‘stigma’ has yet to be conducted. These stigmas significantly impact instances of help-seeking and help-offering by those experiencing and those exposed to people with suicidal thoughts alike [13].

### 1.2. Help-Seeking

FIFO communities are at a greater risk of suicidal behaviours than the general population due to the stigmatisation of help-seeking once suicidal ideation occurs [4,14]. Peer-based training programs can increase help-seeking behaviours on mine-sites; however, research does not specify which attitudes exist that indicate poor help-seeking behaviours prior to training, outside of the broad terminology of “stigma” [15,16,17]. Czyz and colleagues [18] investigated help-seeking attitudes in a sample of at-risk American college students and found that the beliefs that treatment was not needed, a lack of time, and a preference to self-manage were barriers to help-seeking. Such a study has yet to be replicated with FIFO workers.

### 1.3. Help-Offering

Stigma also exists towards help-offering, which limits the likelihood one will intervene in another’s psychological distress [19]. Burnette et al. [19] identified stigmatic barriers to help-offering, including a lack of understanding of suicide, perceptions that those with suicidal ideation are weak, and individuals not believing it is their responsibility to intervene. Burnette and colleagues sampled a military population, obtaining results which, despite the cultural similarities (e.g., isolation and lack of autonomy) [20], are yet to be replicated within a FIFO cohort. The studies mentioned above in relation to help-seeking [15,16,17] also found an increase in help-offering behaviours after the delivery of a peer-support training program; however, they again did not identify what specific attitudes underpinned the stigma that indicated low levels of help-offering in the first instance.

### 1.4. Stigma

Stigma is an inaccurate or negative attitude about a group of people that stems from a lack of education or fear, resulting in ostracization, denial of power [21], and social inequality [22,23]. Mental illness stigma can be deconstructed into three levels. Structural stigma refers to the systems, policies, rules, and structures by which those with mental ill-health are restricted or limited from fully participating in life [22,24,25]. This leads to a disparity of attainability of means between those with mental illness and those without, from which interpersonal stigma is born. Interpersonal stigma (otherwise known as public or social stigma), stems from the consequences of structural stigma (e.g., homelessness, mental illness) and aims to identify the differences between the mentally ill and neurotypicals, assigning labels to the mentally ill [26]. These labels often stem from a lack of education [27]. Stereotypes, such as those where mentally ill people are seen as weak, violent, crazy, not seeking proper treatment, or ‘faking it’, develop into prejudices such as anger and fear, and discriminatory actions such as avoidance, coercion, and violence [27,28].

Self-stigma (otherwise known as intrapersonal stigma) is the acceptance of the public attitudes or stereotypes of mental illness, and the internalisation of those views by members of the stigmatised group [22,26]. For example, the stereotype that depressed people are lazy could be internalised by someone with depression, resulting in their belief that they are simply lazy [26].

It has been argued that suicide stigma is distinct from mental health stigma [27,29,30]. While the sociological structures are the same as mental health stigma, suicide stigma is distinct in that social disapproval, isolation, and exclusion are more overt than mental health stigma [30,31]. In addition, given that not all people who die by suicide have a mental illness [27], examination of the phenomenon of suicide stigma, independent of mental health stigma, is required [32]. Sheehan et al. [32] studied the differences between mental health and suicide stigma, finding that participants believed recovery from depression was more attainable than recovery from a suicide attempt, concluding that social constructs such as religious pressures and shame produced more harmfully stigmatising attitudes towards suicide than mental health.

### 1.5. The Current Study

Although Burnette et al. [19] and Czyz et al. [18] identified some stigmatised attitudes that presented as barriers to help-seeking and help-offering in military college students, respectively, similar studies have not been replicated in the FIFO industry. Given the at-risk status of FIFO workers [5], coupled with a lack of research on the attitudes underpinning suicide stigma in the FIFO industry outside of the broad term ‘masculine attitudes’ [7], a clear knowledge gap exists. The current study utilised an online-self report survey to gain an understanding of the attitudes towards suicide, help-seeking, and help-offering of those working in the FIFO industry to address the following research questions:What are FIFO workers’ attitudes towards suicide?What stigmatised beliefs may hinder FIFO workers from seeking help if they had suicidal thoughts?What stigmatised beliefs might hinder FIFO workers from offering help to a colleague they suspect of having suicidal thoughts?

## 2. Methods

### 2.1. Study Design

A cross-sectional qualitative study design was applied, using a general thematic analysis [33]. The thematic analysis design was considered the most appropriate approach for interpreting the subjective experiences of participants in a way that created an understanding of the collective narrative [34].

### 2.2. Measures

An online survey with open-ended questions was designed to capture qualitative responses from FIFO workers. The survey was created using the online survey software Qualtrics Version 0724, (www.qualtrics.com). Five closed-ended questions were included to obtain demographic information of the participants, followed by seven free-text open-ended questions. The open-ended questions (for example, “What would make you hesitant in seeking help if you were having thought of suicide?”; “What would make you hesitant offering help to someone in the FIFO community if you thought they might be having thoughts of suicide?”) invited participants to express their beliefs on suicide, help-seeking, and help-offering within the FIFO community. At the end of the survey, phone numbers of available support-lines were provided should participants experience any distress while undertaking the survey.

### 2.3. Procedure

Between 13 June 2024 and 14 July 2024, convenience sampling was used to recruit participants who self-identified as current FIFO workers within Western Australia via a QR code and hyperlink that took them to the survey page. The QR code and hyperlink were disseminated via social media (LinkedIn and Facebook), as well as via News Alerts from the Department of Energy, Mines, Industry Regulation and Safety (DEMIRS), WA. Respondents were directed to a landing page that provided them with information about the study, the risks involved, confidentiality and informed consent information, any conflicts of interest that existed, and support service details. Participants were required to consent online before proceeding. Given this sample is from a potentially vulnerable population [8], protective measures such as the provision of support service contact details at the start and end of the survey were included. This study was approved by Griffith University Human Research Ethics Committee (GU reference number 2024/337).

The qualitative nature of the study means that sample size was not a priority [35], and therefore a no a priori sample size requirement was set, with sampling continuing until thematic data saturation was reached (i.e., when no new information or insights emerged from the data) [36].

### 2.4. Participants

A total of 138 (*n* = 138) FIFO workers participated in the study, of which 101 (73%) were males, 37 (27%) females, and none identified as non-binary, prefer not to disclose, or self-described. Ages ranged from 23 to 76 years, with a median age of 39. Of the 138 participants, 96 (70%) identified as having lived or currently living with experience of suicide, with no follow-up qualifiers as to whether this identification related to the self, bereavement, or exposure while supporting another individual exhibiting suicidal behaviours.

### 2.5. Data Analysis

The survey responses were uploaded to the web-based qualitative data analysis software ATLAS.ti 9 (www.atlasti.com). The data were coded using an inductive analysis approach [33], which required familiarisation with the data, generation of codes, search for themes, revision of themes, and finally definition of themes. Themes were not bound to the survey questions, but rather the questions served as a prompt for which the participants could illustrate their thoughts so that significant themes could emerge without being coded to an answer from each question [33].

Per Braun and Clarke (2006) [33], the significance of the themes did not rely exclusively on the prevalence of a theme’s presentation, but rather the significance of the theme, as determined by the research team. As such, quantitative data on theme prevalence were not captured. It must be acknowledged that the themes are derived from the researcher’s interpretation of the answers and the patterns presented. While the inductive analysis is not driven by the researcher’s interest, it is impossible to remove their influence, ideas, and knowledge from the process [33]. The data were initially coded by JJ, which were shared with VR, after which the initial themes were identified. After several iterative discussions and revisions, a consensus was reached upon the final themes and sub-themes. By validation and coding between the two researchers, this methodology sought to mitigate, as much as practicable, the subjectivity of the findings [37].

## 3. Results

Thematic analysis of the data identified four overarching themes related to the participants’ attitudes towards suicide, and stigmatised attitudes that might hinder help-seeking and help-offering:Fear of Negative Consequences for Employment;Lack of Trust in Leadership and Workplace Culture;Perceived Inability to Respond to Suicidal Disclosure;Fear of Negative Reactions.

The themes each comprised between three and four sub-themes. Representative quotes from participants are provided to illustrate each of the sub-themes.

### 3.1. Fear of Negative Consequences for Employment

Fear of Negative Consequences for Employment was identified as the most predominant theme throughout the data, and highlights the participants’ fears of discrimination, the potential negative impacts suicidal disclosure might have on their employment, and the subsequent consequences of that impact.

#### 3.1.1. Fear of Job Loss

Participants expressed a fear that because of their suicidality, their employer would see them as a liability, leading to job loss, as demonstrated in the following quote: “…the idea that [their] job might be at risk if people thought you couldn’t cut it”. Participants reported that they and other workers would rather keep the information to themselves or seek help from an external source than disclose suicidality to their employer and risk losing their job: “…keep a safe distance in order to not offend or get involved and safeguard their job”.

#### 3.1.2. Fitness for Work

Participants also reported that separate to the perceived career-ending implications of suicidal disclosure, they may be declared mentally unfit for work and be stood down. As one participant stated, “…admitting to mental health struggles could jeopardise employment, lead to fewer opportunities, or result in being deemed unfit for work”. They believed that mining companies would declare a worker experiencing suicidality unfit for work and send them offsite, rather than keeping them onsite and having to manage the suicidality and risk a psychosocial injury. “Mining companies don’t want the statistic against them. It also costs them money to manage mental health”.

#### 3.1.3. Negative Financial Impact

Participants spoke of workers’ fears of the impact of being declared unfit for work or that having their employment terminated would also negatively affect their finances, for example, “the thought that [they] may be stood down medically from work and can’t earn money”. Participants described how the loss of income would impact their families and their mental health. “Speaking up may hinder my ability to feed my family. I am told constantly that those that have mental health issues very quickly get removed from the company”.

#### 3.1.4. Fears for Career Progression

Participants also believed that even if they did not lose their job, disclosure would still have an impact on their capacity for career development. The concept of being labelled as “mentally ill” permeated their responses, with many participants describing their fears, such as “fear of ridicule and criticism, and restrictions on career progression”, and believing that that label would have negative ramifications, such as “…long lasting and potential career ending implications”. Participants also expressed concerns that if they offered help to others, they could be reported for being “nosy”. They highlighted that by not interfering in “others’ business”, they were safeguarding their own jobs.

### 3.2. Lack of Trust in Leadership and Workplace Culture

While the previous theme focused specifically on the outcome suicidal disclosure may have on the participants’ employment, this theme describes the systematic and cultural hinderances participants perceived as obstacles to disclosure.

#### 3.2.1. Confidentiality

Participants believed that a lack of confidentiality is a significant deterrent for help-seeking. This was due to their concerns of personal and sensitive information being shared widely and consequently having an impact on their relationships with colleagues. As one participant described, “…personal struggles might become widely known or impact professional relationship…” Some participants disclosed their own or others’ personal experiences where confidentiality of suicidal disclosure was poorly managed, denoting that for some FIFO workers, this was more than just a stigmatised belief, but a reality. One such example was articulated by a worker who described how a colleague’s confidential mental health issues were managed poorly: “…He has since been put on leave and sadly I’ve overhead my boss and his boss openly discussing him at the lunch table!”.

In addition, participants expressed concerns that disclosures to support services would not be treated confidentially and could be discovered by their employers, consequently affecting their employment. “Part of it [is] that they would tell my supervisor, and it would go on my record”. Participants also reported they would be reluctant to disclose suicidality as the lack of confidentiality could lead to gossip within the FIFO community. “I think the majority of people would gossip about it”. “People cannot keep their mouths shut especially about private matters for others”.

#### 3.2.2. Perceptions of Poor Leadership

This sub-theme is driven by a pervasively negative belief expressed by participants that leaders are not empathetic enough, nor adequately trained to appropriately manage disclosures of suicidality. Some examples of comments were “lack of trust, negative response from leadership”, and “the lack of confidentiality and the high number of untrustworthy leaders who typically push people out of the workplace when they raise [their mental health] concerns”.

Participants expressed concerns that disclosing suicidality to leadership would have negative implications for their employment. This belief that employers would be unable to appropriately handle disclosures meant that even those participants who said they would seek help emphasised that they would only do so from an external source. “If I had a Line Manager or Functional Leader who did not show any genuine compassion or respect for their workforce, I would hide it and seek help outside of the organisation”. Some participants conveyed that even if they did trust their immediate leader enough to disclose suicidality, the leader is bound by company policy to escalate the matter, which would have negative consequences for their employment.

#### 3.2.3. Unsupportive Workplace Culture

This sub-theme refers to participants’ beliefs that the culture of the FIFO community is permeated with hypermasculine attitudes, and as such would be unsupportive of a worker that disclosed suicidality. Some participants reported that there are more conversations around mental health than previously. Despite this, it was also reported that the culture on mine-sites is one that promotes stoicism and denounces expressions of vulnerability, with some describing the FIFO culture as toxic: “A harsh culture towards vulnerability on certain sites”. “Possibly the culture on sites would affect the decision as well, some places are very toxic”. “It’s the most depressing, unsupported, bulling, harassment culture I have any been a part of. It’s something I’ve never seen before and doesn’t seem to be changing anytime soon”. Participants believed this toxic culture creates shame and ridicule around mental health, resulting in a significant barrier to help-seeking. “…the culture of the site is one of shame and ridicule for those who suffer with any type of mental health concerns”.

### 3.3. Perceived Inability to Respond to Suicidal Disclosure

When asked for their thoughts about someone who dies by suicide, participants generally expressed empathy and understanding for suicidal people, with many stating that they personally would not hesitate to help a colleague in suicidal distress. However, they also highlighted that they did not feel they had the ability to respond to colleagues’ disclosures of suicidality and were fearful of the consequences if they offered help anyway.

#### 3.3.1. Lack of Knowledge and Confidence

Despite empathy and a willingness to offer help, participants felt that they lacked knowledge on how to do so. “I would want to help someone, but I would not know how or the signs that they needed urgent help or just a chat”. This lack of knowledge meant that participants also lacked confidence in their ability to intervene, with some expressing that engaging in help-offering was beyond their own or their colleagues’ ability. “They also might feel like they are not equipped to handle those kinds of conversations—they lack the confidence in their ability to help”. “… involvement in something like offering help to someone with suicidal thoughts might be beyond [their] ability”. Some participants described how they lacked knowledge of appropriate services to which they could refer a suicidal colleague: “…not knowing information or who to send the people to get help”. In contrast, other participants perceived that there has been an increase in available services onsite: “There are a number of internal and external services available to help people, if they are willing to be vulnerable and open to help”.

#### 3.3.2. Fear of Saying or Doing the Wrong Thing

While the previous sub-theme identified a lack of knowledge and confidence, this and the following sub-themes identify fears of engaging with a suicidal colleague without the appropriate knowledge. Participants felt that they would be worried about saying or doing something that would make the situation worse. As one participant stated, this belief stems from the idea that “…the wrong language or approach to someone in apparent crisis could be the trigger”. Participants highlighted that “… talking to them might do more harm than good”. This fear that they would motivate someone towards suicidal behaviour was a notable deterrent to help-offering: “I am never one who knows what to say to a person and I would be scared that I would say the wrong thing. I am not qualified to deal with that”. “They might make the situation worse and won’t be able to help anyway”.

#### 3.3.3. Guilt from Not Being Able to Help

Similarly, not being able to help someone experiencing suicidality and subsequent guilt was a significant fear for participants. Some felt that if they were to get involved, their inability to effectively help would mean the person would suicide anyway: “… knowing I may not be able to provide the right information/advice/care to prevent them from going ahead”. “Them not listening or you not being able to stop [prevent the suicide]”. Participants highlighted that if their intervention was ineffective and the person died by suicide, they would feel guilty and responsible for that person’s passing. “I would feel I’m not qualified to deal with it and if something happened, and they ended it I would feel survivors’ guilt”. The concept of feeling guilty about being unable to effectively intervene and prevent a suicide was a major deterrent for participants in offering help to a coworker who may be suicidal.

#### 3.3.4. Fear of Becoming Responsible for Recovery

Participants also expressed a fear of becoming responsible for the recovery of the person if they did offer help to a suicidal coworker. Some disclosed their lived experiences of instances where there was an emotional toll of feeling responsible for helping, “… [it] burnt me out to the point that I no longer speak to that person”. Others were concerned that their support would be required for an extended period, or that this would “…add… others’ problems to their own”. This highlights the participants’ perceptions that they do not have the knowledge to appropriately manage the disclosure of suicidality, and as a result would choose not to offer help. “I don’t want that responsibility. If I was close enough to them to see they needed help, then they would have to be a friend. I know I should, but I am not qualified”.

### 3.4. Fear of Negative Reactions

This theme identifies a fear that the person seeking or offering help may encounter an acute negative reaction.

#### 3.4.1. Fear of Violence and Aggression

Participants highlighted their concerns of an aggressive reaction from a person to whom they offered help. Some participants claimed they feared that they would be met with physical violence from the person as well as more prolonged “retribution”-style aggression. Consequently, fear for their personal safety was reported as a notable hinderance for them in engaging in help-offering behaviours.

#### 3.4.2. Negative Impact on Relationship

Participants were also concerned that the offer of help to a coworker would negatively impact their relationship. They highlighted that by attempting to offer help, they might inadvertently “overstep” boundaries, resulting in a strain on or complete breakdown of the relationship. They also worried that the person may deny suicidality, causing offence, frustration, and the person distancing themselves from the help-offeror. This sentiment extended to professional relationships as well, and the impact it could have on their entire team. “In a work environment like FIFO, where teamwork and collaboration are crucial, I might be concerned that addressing such a sensitive issue could impact my professional relationships or team dynamics”.

#### 3.4.3. Fear of Rejection

Concern that the help-offeror may be rejected was also identified. In addition to a general fear of rejection, participants highlighted that the rejection may be “hostile” and the language used towards the help-offeror aggressive, such as “Being told to mind their own business”. This concern of rejection was identified as a key reason participants would not offer help.

#### 3.4.4. Fear of Being Mocked

Where the previous sub-themes are illustrative of the concerns of the help-offeror, this one stemmed from the perspectives of the help-seeker. Participants reported their fears of being mocked, for example, “Being laughed at and told to toughen up”. Another expressed concerns about being treated differently: “I don’t want anyone feeling bad for me or treating me differently because of how I feel”. Adjectives such as “weak”, “gutless”, and “soft” were used by participants to describe the anticipated reaction from other workers if they sought help. They reported that they would not seek help from others in the industry due to the perception that they would be mocked and not supported.

## 4. Discussion

The current study aimed to qualitatively examine the attitudes towards suicide, help-seeking, and help-offering that underpin suicide stigma in the FIFO industry. To the best of the authors’ knowledge, this is the first study to do so. General thematic analysis identified four key themes: Fear of Negative Consequences for Employment, Lack of Trust in Leadership and Workplace Culture, Perceived Inability to Respond to Suicidal Disclosure, and Fear of Negative Reactions. Based on these themes and their sub-themes, the research team was able to draw conclusions in relation to all three research questions. It should be noted that while some divergent views were present in the participant responses, they did not meet the criteria to form individual themes or sub-themes. However, in the discourse on workplace safety revision, it is important to ensure that divergent views are included in the conversation, as exclusion of these perspectives can serve to increase stigmatised attitudes amongst the disillusioned [38].

### 4.1. Attitudes Towards Suicide

Overall, the participants expressed empathy towards those that had died by suicide or were experiencing suicidality, despite being fearful of seeking and offering help. This finding is not consistent with previous research that indicates workers in male-dominated environments typically perceive those that die by suicide as weak or failures [12]. However, in contrast to this empathy they expressed, the participants believed that other workers in the FIFO industry would hold the above-mentioned stigmatised attitudes towards them if they were to seek help. There are two potential explanations. Firstly, the majority of the sample (70% of participants) identified as having a lived or living experience of suicidality. It has been established that those with lived experience are naturally more empathetic towards others with suicidality [39]. Therefore, it is probable that the participant cohort may be more empathetic than the general population of FIFO workers. The second possible explanation is that despite the empathy expressed by participants, the suicide stigma that permeates the industry is such that participants may incorrectly believe that others are less empathetic than themselves. Further research is required to determine if the empathetic attitudes towards suicide are representative of the broader FIFO community.

### 4.2. Attitudes Towards Help-Seeking

The second research question sought to understand what stigmatising attitudes may hinder good help-seeking practice in the FIFO industry. The most salient finding of the current study is that many participants would not seek help for suicidality due to a fear of negative ramifications for their employment. Participants reported they believed a disclosure of suicidal ideation or behaviour to their employer would negatively impact not only their current job, but their reputation and thus chances for career progression. This fear stemmed from several beliefs including that they would be declared unfit for work, would lose financial stability, would be sacked, or would be singled out as a liability.

The theme of negative ramifications for employment also extended to concerns regarding confidentiality. Participants were fearful that their disclosure would be shared amongst their colleagues, citing concerns of being judged, ostracised, and the subject of gossip. This is not an unreasonable concern, given that some participants reported firsthand experiences of their own and others’ mental health disclosures being openly discussed and mocked by leaders in common workplace areas. The concept of being seen as separate and alienated from their colleagues due to disclosures of suicidality aligns closely with the concept of “thwarted belongingness”, which, according to Joiner’s Interpersonal Theory of Suicide [40], could increase a sense of isolation and potentially further contribute to suicidality. Perceptions of thwarted belongingness within the community where one spends most of their time could have a significant impact on workers’ capacity to function and seek further help, particularly in the unique FIFO working environment. It is critical that FIFO organisations address their capacity for appropriately managing confidential disclosures of suicidality, and in doing so effectively communicate these improvements to ensure workers are confident in the systems they would rely on in such cases.

Participants’ concerns about being declared unfit for work may stem from recent changes to the legislation that governs WA workplaces. The Work Health and Safety Act 2020 (WA, AUS) (Act) [41] dictates that an employer (Person Conducting Business or Undertaking or PCBU) must ensure as far as is reasonably practicable the health and safety of workers while they are at work. Supporting the Act is the Work Health and Safety (Mines) Regulations 2022 (WA, AUS) (Regulations) [42], which states that this duty to ensure health extends to psychosocial health. This means that any mental ill-health that a worker experiences while at work, which is caused by work-related factors, can be considered the obligation and failing of the employer [41,42]. This also extends to the safety of other workers who may be put at risk through the actions of someone that is under psychological distress; for example, worker A is struck by a vehicle that was driven by worker B, while worker B was fatigued due to late night suicidal ruminations.

Under the Act, the fine for failing to provide and maintain a safe working environment, which causes the death of, or serious harm to, a worker, is 3.5 million dollars (AUD), [41]. Being found guilty of failing to maintain a safe workplace, resulting in the suicide of a worker, would not only have financial penalties but also cause reputational damage to the organisation. It is argued that the threat of these consequences would cause significant concern for any organisation operating under the Act and Regulations. Because the Act states that the employer is responsible for a worker’s safety while they are at work [41], it can be understood why declaring a worker who has disclosed suicidality as unfit for work and removing them from the workplace could be a quick and preferable action for the employer to mitigate risk to themselves, the worker, and their colleagues. However, as the current study shows, the fear of being declared unfit from work and being removed from site is a significant barrier to help-seeking, consequently minimising the chance for intervention and increasing the likelihood of suicide. Further research is required to better understand the significance and extent that employers’ concerns around the fitness for work of employees have on workers’ disclosure of mental health issues and suicidality, as well as the impact that the legislation has on a PCBU’s attitudes and actions towards mental health in the workplace.

In addition to a lack of trustworthiness and a fear of negative ramifications for the help-seekers’ employment, participants reported they believed their leaders were ill equipped to appropriately handle disclosures of suicidal ideation or behaviour. This was expressed as concerns about their leaders’ lack of training on the matter, lack of empathy, and that their leader would be bound by company policy and report the matter upline, having negative ramifications for their employment. This presents an opportunity for the FIFO industry to provide evidence-based training to their frontline, senior, and HR leaders on how to appropriately manage disclosures of suicidal ideation and behaviour. This recommendation should be met with caution; however, adequate training alone will not address the stigmatised attitudes towards leadership, nor increase positive help-seeking behaviours if workers are too fearful to engage.

Participants noted an increase in awareness of available support services. This, however, was also met with a somewhat contrary position that while there was some awareness of support services, they were not necessarily trusted and information on how to access the services was lacking. These findings are consistent with a qualitative study of Australian male blue- and white-collar workers which found that while workers were aware of Employee Assistance Programs, there was low uptake due to lack of trustworthiness and confidentiality, and a fear of stigma and career jeopardy [43]. Many participants in the current study reported that they would rather seek help from an external source than seek help within their organisation’s available support systems. The trepidation in engaging with support services in the current study extended to seeking support from within the participants’ organisational structures. It could be argued that having support services available within the FIFO community is not an effective intervention if workers are too fearful to engage with them.

### 4.3. Attitudes Towards Help-Offering

The final research question aimed to identify the attitudes held by FIFO workers that would hinder them from offering help to a colleague they thought to be experiencing suicidality. Despite having general empathy, participants reported great hesitation in offering help to someone they suspected to be suicidal. Participants felt they lacked the confidence and knowledge to effectively intervene. This was driven primarily by fear that their actions could worsen the person’s suicidality, with some participants fearful of the guilt they would feel if they failed to prevent a coworker’s suicide. This is consistent with the current understanding of suicide stigma [44]. People often incorrectly believe that talking about suicide is likely to make suicidal behaviours worse, and that saying or doing the wrong thing can lead to suicide [44], with males being identified as most likely to believe this [45]. A multitude of evidence exists that shows these myths are untrue, and that education on the matter is key to debunking them [45].

A lack of suicide literacy in males is a previously identified problem [29,46], with this poor literacy by extension permeating male-dominated industries [13]. Several organisations such as ‘Mates in Mining’ exist, providing training programs to educate and de-stigmatise suicide in male-dominated industries [47,48]. Research has shown a significant self-identified increase in suicide prevention literacy in mining workers after the delivery of the Mates in Mining training [47]. However, in light of the current research findings that FIFO workers lack the knowledge and confidence to offer help, coupled with the stigmatised attitudes that they may say or do something to make it worse, it is clear that education and de-stigmatisation training such as the ‘Mates in Mining’ program need to continue being provided to FIFO workers at a greater rate than they currently are.

The fear of negative consequences for employment again presented as a barrier to positive help-offering behaviours. Participants also expressed a concern that they could be subject to disciplinary action if the person to whom they offered help took offence and reported them to management. This reflects a concerning perception of the FIFO culture, where self-preservation is favoured over offering necessary peer support. It is recommended that government and FIFO organisations develop and promote safeguards for help-offerors to encourage positive help-offering behaviours.

Participants also reported concerns about violence and aggression from the subject as a barrier to help-offering. This aligns with current understanding on stigmatised attitudes towards mental health, specifically the belief that mentally ill people are violent, unpredictable, or aggressive [28]. This finding also supports the need for further psychoeducation on responding to mental ill-health. However, several participants reported that they were concerned a violent or aggressive reaction would present if a coworker took offence to the assumption that they were experiencing suicidality. This may be explained by the current understanding of hyper-masculine attitudes that permeate male-dominated industries [12]. However, further research is warranted to better understand these attitudes.

### 4.4. Stigma

Participants reported they were fearful of losing their jobs, fearful of being the subject of gossip, fearful of making a situation worse, and fearful of having the subject react negatively if they offered help. This finding is consistent with the current literature on suicide stigma that indicates fear is a primary driver of public stigma, which, in turn, individuals internalise as their own beliefs [30]. Individuals become fearful of the negative reactions of others, based upon the stigmatised beliefs of the collective community [30]. Based upon this theory, we can extrapolate that the fears expressed by the individual participants in the current study (intrapersonal stigma) are representative of the fears and stigmatised attitudes of the broader community (interpersonal stigma), driven by the policies, procedures, and available services that manage mental health (structural stigma) [22].

The current results are consistent with Javed et al.’s model of stigmatisation [22], which could be adapted to the FIFO setting. While structural stigma traditionally considers the social and political structures that govern mental health [22], in the context of this research it refers to the structures within the workplace. Workplace policy that governs how a worker experiencing suicidality is managed (e.g., HR procedures, fitness-for-work models, and mental health transportation guidelines), coupled with the Act and Regulations that legislate a PCBU’s duty to provide psychosocial safety to its workers, appear to contribute to the creation of suicide stigma.

Considered in the context of the current research, interpersonal stigma [22] means that the structurally motivated stigmas, such as declaring a worker unfit for work if they are suicidal, aid in the formation of the broader FIFO community’s views on suicide. This can be seen in the fears participants expressed about being viewed as “weak”, as not being fit for work means one is not fit for the FIFO environment. This also extends to the views on the untrustworthiness of leaders, who participants believed would lack empathy or care regarding disclosure of suicidal behaviours. Participants also reported that the nature of FIFO is such that their colleagues form part of their onsite community. In turn, any disclosure of suicidality that is not treated in confidence by management would likely be shared amongst the community members where they may be judged. This fear of public judgment and ridicule that participants expressed reflects the current understanding of interpersonal stigma, whereby people do not want to be “othered” by their peers by virtue of being considered mentally ill or “weak” and therefore avoid help-seeking behaviours altogether [26].

With regard to the current study, intrapersonal stigma can be interpreted as the stigmatised attitudes held by the collective (e.g., a person will lose their job if they disclose suicidal behaviours) that are internalised and believed by those experiencing suicidality (e.g., I will lose my job if I disclose my suicidal behaviours). It is not possible to ascertain if these beliefs were held by participants experiencing suicidal behaviours in the current study, as specific information on current suicidal behaviours was not obtained. However, it can be concluded that the intrapersonal stigmatised attitudes expressed in this study appear to hinder a FIFO worker experiencing suicidality from seeking help, and limit someone who suspects a colleague of experiencing suicidal behaviours from offering help.

### 4.5. Strengths and Limitations

This research extends the current body of knowledge by providing valuable insights into the attitudes that underpin suicide stigma in the FIFO industry. It should be considered as a preliminary examination of these attitudes, and as such has a number of limitations. A major strength is the application of a qualitative approach to an industry online survey to assist in capturing rich and detailed data. Convenience sampling enabled the collection of data from a sizable sample of FIFO workers, which would not have otherwise been achievable due to resourcing limitations. However, the use of convenience sampling comes with several inherent limitations. Firstly, participation bias is possible, meaning those interested in the study or with a lived experience of suicidality may be more interested in participating [49]. The voluntary online self-report nature of the study means that participant intention and reasoning for participation were unexamined. However, the current literature on suicide considers that the involvement of participants with lived experience in research is critical in ensuring informed and relevant data are reported [50]. Further to this, the online self-report nature of the survey meant that participant information could not be verified. It is therefore possible that despite recruitment being targeted at WA FIFO workers, other people outside the expected participant pool may have participated. In addition, the cross-sectional nature of the study limits the capacity for causal inferences. Future research should consider a more robust sampling method to increase participation of those without a lived experience of suicide to gain more representative information. The current study examined themes across all participants, without consideration of demographic differences (e.g., gender, age, or lived experience). Future research should aim to examine the between-group differences for the different demographics to gain a more in-depth understanding of this phenomenon.

## 5. Conclusions

The findings from this study present new and valuable insights into the reasons FIFO workers are reluctant to offer or seek help. This study does not contradict previous findings but rather provides a deeper understanding of stigma structures in the FIFO work environment. While some limitations exist in relation to the sampling method and data collection, the method was considered the most appropriate given the limitations on researcher resources and funding. This study has implications for non-clinical intervention practices. Namely, FIFO organisations need to urgently address the systematic inadequacies that exist within their company structure that may hinder disclosure of suicidal ideation, as well as increase suicide education and de-stigmatisation initiatives available onsite. Governments should consider the implication of recently enacted legislation that may influence discriminatory practices by workplaces, as well as provide support to FIFO organisations in delivering suicide education and de-stigmatisation programs.

## Data Availability

Data are not available due to ethical and confidentiality issues associated with qualitative data.

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
