# Peer review of "Understanding Suicide Stigma in Fly-In/Fly-Out Workers: A Thematic Analysis of Attitudes Towards Suicide, Help-Seeking and Help-Offering"

_ijerph, 2025, doi:10.3390/ijerph22030395_

Round 1

Reviewer 1 Report

Comments and Suggestions for Authors

Summary

This paper presents a qualitative study examining attitudes towards suicide, help-seeking, and help-offering among fly-in/fly-out (FIFO) workers in the Australian mining industry. Using thematic analysis of survey responses from 138 FIFO workers, the study identified four main themes related to stigma and barriers. The research provides valuable insights into the factors underlying suicide stigma in this high-risk population and has important implications for addressing systemic issues hindering disclosure and access to support.

General Comments

Strengths:

  • The qualitative approach is appropriate for exploring complex attitudes and experiences in this understudied population.
  • The sample size (n=138) is substantial for a qualitative study, allowing for data saturation.
  • The four main themes identified provide important insights into barriers to help-seeking and help-offering behaviours.
  • The findings are well-contextualized within existing literature on suicide stigma.
  • Implications for policy and practice are clearly articulated.

Weaknesses:

  • More details on the data analysis process, including inter-rater reliability, would strengthen methodological rigor.
  • Quantitative data on theme prevalence would provide additional context.
  • Deeper discussion of how structural factors interact with individual attitudes is warranted.
  • The cross-sectional design limits causal inferences, which should be more explicitly acknowledged.

Specific Comments

  1. Introduction: The literature review effectively establishes the context and rationale for the study. Consider adding more recent statistics on suicide rates in the mining industry, if available.
  2. Methods:
    • Provide more detail on how thematic saturation was determined.
    • Clarify the rationale for using convenience sampling and discuss potential limitations.
    • Include information on inter-coder reliability measures.
  3. Results:
    • Consider presenting a table summarizing the themes and sub-themes with frequency data.
    • Provide more context around participant quotes to enhance thick description.
    • Include any negative cases or divergent views to enhance analytical depth.
  4. Discussion:
    • More critically analyse how findings compare to previous studies on suicide stigma in male-dominated industries.
    • Expand on the implications of recent workplace health and safety legislation for managing suicide risk in FIFO settings.
    • Discuss potential strategies for addressing the identified barriers at structural and organizational levels.
  5. Limitations:
    • Acknowledge potential participation bias due to convenience sampling.
    • Discuss limitations of self-report data and cross-sectional design more explicitly.
  6. Ethics:
    • Provide more detail on ethical considerations specific to conducting suicide research with a vulnerable population.
  7. References:
    • Ensure all cited works are recent (within 5 years) where possible, particularly for statistics and policy information.

Author Response

1. Summary

Thank you very much for taking the time to review this manuscript. Please find the detailed responses below and the corresponding revisions in the re-submitted files.

2. Point-by-point response to Comments and Suggestions for Authors

Comments 1: More details on the data analysis process, including inter-rater reliability, would strengthen methodological rigor.

Response 1: We thank the reviewer for this feedback. We have now added further detail on the data analysis process, with reference to established thematic analysis methodology. However, as guided by the approach of Braun and Clark (2013) who assert that reliability is not an appropriate criterion for judging qualitative work and may be epistemologically problematic in qualitative analysis, we did not conduct quantitative inter-rater reliability testing (i.e., the kappa statistic). Rather, we relied on an iterative validation process to achieve consensus between the two researchers. Please see our revisions in section 2.5. Data Analysis. We hope that this has satisfied your valid query.

Comments 2: Quantitative data on theme prevalence would provide additional context.

Response 2: Thank you for your comment. The Braun and Clarke (2006) methodology that we applied discusses that quantitative measures of theme prevalence are unnecessary and sometimes a hindrance. As such, we respectfully will not be adding this information. However, to your point, we have explicitly added this context to the manuscript (with the relevant reference) to provide additional clarification (see Lines 171-174). Thank you for identifying this and providing the opportunity for us to provide further clarification.  

Comments 3: Deeper discussion of how structural factors interact with individual attitudes is warranted.

Response 3: Thank you for this feedback. We believe these interactions have been addressed in the discussion section. Please see pages 9-12.

Comments 4: The cross-sectional design limits causal inferences, which should be more explicitly acknowledged.

Response 4: Thank you for pointing this out. We agree, and have amended section 4.5 Strengths and Limitations, (Lines 572-573), to explicitly acknowledge this.

Comments 5: Consider adding more recent statistics on suicide rates in the mining industry, if available.

Response 5: Thank you for pointing this out. To our knowledge, the 2023 study by King et al. is the most recent dataset available on suicide rates in the mining industry. We are aware of some research currently being conducted by our colleagues, however the findings have yet to be published.

Comments 6: Provide more detail on how thematic saturation was determined.

Response 5: Thank you for this comment. We have therefore added additional information in section 2.3. Procedure, (Lines 152-155).

Comments 7: Clarify the rationale for using convenience sampling and discuss potential limitations.

Response 7: Thank you for your comment. We have provided further rationale for our use of convenience sampling and added additional limitations to this methodology. Please see our revisions in s. 4.5 Strengths and Limitations (Lines 563-580).

Comments 8: Include information on inter-coder reliability measures..

Response 8: Thank you for your comment. Please see Response 1 for our reply to this point.

Comments 9: Consider presenting a table summarizing the themes and sub-themes with frequency data.

Response 9: Thank you for your comment. As discussed in Response 2, we followed the Braun & Clarke (2006) methodology that cautions against the use of quantitative values such as frequency data to validate the creation of themes and sub-themes. As such, we respectfully decline to add such a table, however, have added explicit clarification on this as seen in Response 2.

Comments 10: Provide more context around participant quotes to enhance thick description.

Response 10: We thank the reviewer for this comment. We believe we have provided considerable context around the participant quotes; however, we have now added further text through sections 3.1 – 3.4 to denote more clearly that the quotes are examples of specific issues. We hope this addresses these concerns.

Comments 11: Include any negative cases or divergent views to enhance analytical depth.

Response 11: We thank the reviewer for their comment and insight. Divergent views did not meet the threshold for inclusion as themes or sub themes. However, to your point, we have added this information for context in the Discussion section (Lines 379-383), as well as a statement on the importance of including these voices in the discussion around workplace mental health.

Comments 12: More critically analyse how findings compare to previous studies on suicide stigma in male-dominated industries.

Response 12: We thank the reviewer for this comment. We believe that we have added adequate comparison between these and previous findings throughout the discussion section. In addition, we have considered possible reasons for variation when these findings were inconsistent with previous ones (see specifically section 4.1 Attitudes Towards Suicide). We hope this answer addresses your concerns.

Comments 12: Expand on the implications of recent workplace health and safety legislation for managing suicide risk in FIFO settings.

Response 12: Thankyou for pointing this out. We agree and have expanded on the potential implications of the legislation. Please see s. 4.2 Attitudes Towards Help Seeking, (Lines 425-452) for our additions.

Comments 13: Discuss potential strategies for addressing the identified barriers at structural and organizational levels.

Response 13: We thank the reviewer for their comment. While we have included some suggestions for areas of improvement (such as reviewing policies that may be discriminatory and continued stigma-reduction education for workers), we believe that it is outside the scope of this research to suggest additional strategies for addressing organizational levels, as we are not subject matter experts on workplace safety reform. As such, we respectfully decline to add any further recommendations to this point, and hope the reviewer appreciates our perspective.

Comments 13: Acknowledge potential participation bias due to convenience sampling & discuss limitations of self-report data and cross-sectional design more explicitly

Response 13: Thankyou for your comment. We agree and as discussed in Response 7, have added additional information regarding the limitations of self-report data and cross-sectional design more explicitly.

Comments 14: Provide more detail on ethical considerations specific to conducting suicide research with a vulnerable population.

Response 14: Thankyou for your comment. We agree this needs further clarification. As such, we have included additional information in s.2.3 Procedure, (Lines 148-150), that speaks to the considerations of ethical engagement with potentially vulnerable participants.

Comments 15: Ensure all cited works are recent (within 5 years) where possible, particularly for statistics and policy information.

Response 15: Thankyou for your comment. Where possible we have included references to research that was conducted within 5 years. However, given the scarcity of research on mining suicide stigma, and lack of statistical data on suicide, (given the complications of confirming a death by suicide), it was not always possible to cite ‘new’ research. In addition, foundational concepts such as thematic analysis methodology and the Interpersonal Theory of Suicide are older than 5 years; however they are enduring theories that we believe are important to cite, rather than recent adaptations. However, to your point, we have reviewed the reference list and (where possible), updated some of the references to reflect more recent data. Specifically, [3], [23], [28], and [45]

Additional clarifications

Thank you again for your time and valued feedback.

Reviewer 2 Report

Comments and Suggestions for Authors

I would like to congratulate the authors on a well-written and interesting manuscript. Nevertheless, I want to address some minor points and make a few more recommendations. I will address these points in accordance with the structure of the present manuscript, namely: Introduction, Methods, Results, Discussion, and Conclusions. Lastly, I will mention a few overall comments that apply throughout the entire work (e.g., grammatical errors). 

Introduction 

More recent studies must be included in the literature review section to expand the discussion and demonstrate the manuscript's relevance. For example, only 1 reference is from 2024, and 6 from 2023 out of 48 references in total, more recent literature could be included. Suggestions:

-       LaMontagne, A. D., Lockwood, C., Mackinnon, A., Henry, D., Cox, L., Hall, N. R., & King, T. L. (2025). MATES in Manufacturing: A Cluster RCT Evaluation of a Workplace Suicide Prevention Program. American Journal of Industrial Medicine. Advance online publication. https://doi.org/10.1002/ajim.23698

-       Wittenhagen, L., Gullestrup, J., Doran, C. M., Brimelow, R., Thompson, N., Heffernan, E., & Meurk, C. S. (2024). The concept of distress – widely used but what does it mean for individuals working in the construction industry? Journal of Workplace Behavioral Health, 1–19. https://doi.org/10.1080/15555240.2024.2356799

Methods  

The methodological design seems to be adequate and based on a solid theoretical and conceptual foundation. Each section's organization includes an adequate rationale.

Results 

The results are clearly presented, and the discussion is elaborated upon, demonstrating critical thinking in light of the findings.

Discussion and Conclusions 

The discussion and conclusions are supported by the results. I did not find any use of jargon, although I did find some grammatical errors, as mentioned in the “overall comments” section below. The manuscript was written in scientific language throughout. 

Overall comments:

Please double-check for grammatical errors or errors, for example:

Abstract, line 20: “help-seeking and help-offering behaviors ty.”, the “ty” at the end seems to be an error.

Line 73, where it reads: “cultural similarities, (e.g., isolation and lack of autonomy) [20] has yet”, should be read “cultural similarities (e.g., isolation and lack of autonomy) [20], has yet”.

Line 571, where it reads FFO, maybe the authors wanted to write FIFO? Otherwise, please clarify the meaning of FFO, as it was not mentioned anywhere else in the manuscript.

Additionally, double-check the references list and standardize the citation format as recommended by IJERPH guidelines. For example, references 6, 12 and others present the article title in capital letters, but the majority of the references do not; also, references as for example 40 and 41, should be accompanied by a link to the respective online documents, if applicable.

Author Response

Responses to Reviewer 2

1. Summary

2. Point-by-point response to Comments and Suggestions for Authors

Comments 1: More recent studies must be included in the literature review section to expand the discussion and demonstrate the manuscript's relevance.

Response 1: Thank you for pointing this out. Where possible we have included references to research that was conducted within 5 years. However, given the scarcity of research on mining suicide stigma, and lack of statistical data on suicide, (given the complications of confirming a death by suicide), it was not always possible to reference ‘new’ research. In addition, foundational concepts such as thematic analysis methodology and the Interpersonal Theory of Suicide are older than 5 years; however, they are enduring theories that we believe are important to cite, rather than recent adaptations. However, to your point, we have reviewed the reference list and (where possible), updated some of the references to reflect more recent data. Specifically, [3], [23], [28], and [45], and included the La Montagne et al., (2025) reference you kindly provided [49].

Comments 2: Please double-check for grammatical errors or errors

Response 2:  Thank you for pointing this out. We have conducted an additional spelling and grammar check, and amended a number of mistakes, including the ones to which you kindly alerted us.

Comments 2: double-check the references list and standardize the citation format as recommended by IJERPH guidelines.

Response 2:  

We have now checked the reference list and amended the citation format so that it is standardized as per the IJERPH guidelines. We have also reviewed the references for standard formatting, an in addition to the examples you provided, have identified others that lack a url link, and added those where possible.

Reviewer 3 Report

Comments and Suggestions for Authors

The authors investigated the attitudes toward suicide, help-seeking, and help-offering of those working in the FIFO (fly in/fly out) industry. For this purpose, they used an online self-report survey. This is certainly a relevant topic and the discussions are interesting. However, the main problem with this manuscript is that it lacks scientific soundness. The authors used an online self-report survey in which participants self-reported to be FIFO workers. However, there is no guarantee of this. Moreover, the only epidemiological information available is the participants' gender, as only the minimum and maximum age range is reported. More stratification by gender and age of participants would have been helpful. In addition, the duration of the participants' work in the FIFO industry would also be crucial. The authors could also include tables and figures. In the results, lines 183-191 are a duplication of what was written just before (lines 174-182). This paper has serious limitations that should be well pointed out if the authors have no way to reduce them. More space should also have been given to discuss what solutions could be introduced for preventive and therapeutic purposes. 

Author Response

Responses to Reviewer 3

1. Summary

2. Point-by-point response to Comments and Suggestions for Authors

Comments 1: The authors used an online self-report survey in which participants self-reported to be FIFO workers. However, there is no guarantee of this.

Response 1: Thank you for pointing this out. We agree, and have amended section 4.5 Strengths and Limitations, (Lines 584-587), to explicitly acknowledge this limitation.

Comments 2: Moreover, the only epidemiological information available is the participants' gender, as only the minimum and maximum age range is reported. More stratification by gender and age of participants would have been helpful.

Response 2:  Thank you for this comment. We note that the median age is also provided, however, agree with your point that further analysis of themes between gender and age groups could be warranted. While it has not been included in the current study, we have added this point to the discussion in the limitations section (Lines 580-584).

Comments 3: The authors could also include tables and figures.

Response 3:  Thank you for your suggestion. Unfortunately, we are unable to infer exactly what data you would like to see presented in the suggested tables and figures. We are mindful of space limitations of the journal, but would be happy to provide any specific tables or figures if requested by the editor.

Comments 4:  In the results, lines 183-191 are a duplication of what was written just before (lines 174-182).

Response 4:  Thank you for bringing this to our attention. We believe this may have occurred while formatting the manuscript into the IJERPH template. The duplication has been deleted.

Comments 5:  This paper has serious limitations that should be well pointed out if the authors have no way to reduce them.

Response 5:  Thank you for your feedback. We agree and have amended the Strengths and Limitations section to better reflect the ‘preliminary examination’ nature of the paper and articulate the limitations in greater depth.

Comments 6:  More space should also have been given to discuss what solutions could be introduced for preventive and therapeutic purposes.

Response 6:  We thank the reviewer for their comment. While we have included some suggestions for areas of improvement (such as reviewing policies that may be discriminatory and continued stigma-reduction education for workers), we believe that it is outside the scope of this research to suggest additional strategies for addressing organizational levels, as we are not subject matter experts on workplace safety reform. As such, we respectfully have not added any further recommendations to this point, and hope the reviewer appreciates our perspective.

Round 2

Reviewer 3 Report

Comments and Suggestions for Authors

The authors have amended the paper according to the comments. Where they were not able to implement the paper, it was highlighted as a limitation in the discussions. I have no further comments.